# Fatty Acid Synthase Is a Key Enabler for Endocrine Resistance in Heregulin-Overexpressing Luminal B-Like Breast Cancer

**DOI:** 10.3390/ijms21207661

**Published:** 2020-10-16

**Authors:** Javier A. Menendez, Inderjit Mehmi, Adriana Papadimitropoulou, Travis Vander Steen, Elisabet Cuyàs, Sara Verdura, Ingrid Espinoza, Luciano Vellon, Ella Atlas, Ruth Lupu

**Affiliations:** 1Program Against Cancer Therapeutic Resistance (ProCURE), Metabolism and Cancer Group, Catalan Institute of Oncology, 17007 Girona, Spain; ecuyas@idibgi.org (E.C.); sverdura@idibgi.org (S.V.); 2Girona Biomedical Research Institute (IDIBGI), 17190 Girona, Spain; 3The Angeles Clinic and Research Institute, Cedar Sinai affiliate, Los Angeles, CA 90025, USA; imehmi@theangelesclinic.org; 4Center of Basic Research, Biomedical Research Foundation of the Academy of Athens, 11527 Athens, Greece; adapapadim@gmail.com; 5Mayo Clinic, Division of Experimental Pathology, Department of Laboratory Medicine and Pathology, Rochester, MN 55905, USA; VanderSteen.Travis@mayo.edu; 6School of Population Health, University of Mississippi Medical Center, Jackson, MS 39216, USA; iespinoza@umc.edu; 7Cancer Institute, School of Medicine, University of Mississippi Medical Center, Jackson, MS 39216, USA; 8Stem Cells Laboratory, Institute of Biology and Experimental Medicine (IBYME-CONICET), Buenos Aires C1428ADN, Argentina; luciano.vellon@ibyme.conicet.gov.ar; 9Environmental Health Science and Research Bureau, Health Canada, Ottawa, ON K1A 0K9, Canada; ella.atlas@canada.ca; 10Department of Biochemistry, Microbiology, and Immunology, University of Ottawa, Ottawa, ON K1N 6N5, Canada; 11Mayo Clinic Minnesota, Department of Biochemistry and Molecular Biology Laboratory, Rochester, MN 55905, USA; 12Mayo Clinic Cancer Center, Rochester, MN 55905, USA

**Keywords:** luminal, tamoxifen, fulvestrant, endocrine resistance

## Abstract

HER2 transactivation by the HER3 ligand heregulin (HRG) promotes an endocrine-resistant phenotype in the estrogen receptor-positive (ER+) luminal-B subtype of breast cancer. The underlying biological mechanisms that link them are, however, incompletely understood. Here, we evaluated the putative role of the lipogenic enzyme fatty acid synthase (FASN) as a major cause of HRG-driven endocrine resistance in ER+/HER2-negative breast cancer cells. MCF-7 cells engineered to stably overexpress HRG (MCF-7/HRG), an in vitro model of tamoxifen/fulvestrant-resistant luminal B-like breast cancer, showed a pronounced up-regulation of *FASN* gene/FASN protein expression. Autocrine HRG up-regulated FASN expression via HER2 transactivation and downstream activation of PI-3K/AKT and MAPK-ERK1/2 signaling pathways. The HRG-driven FASN-overexpressing phenotype was fully prevented in MCF-7 cells expressing a structural deletion mutant of HRG that is sequestered in a cellular compartment and lacks the ability to promote endocrine-resistance in an autocrine manner. Pharmacological inhibition of FASN activity blocked the estradiol-independent and tamoxifen/fulvestrant-refractory ability of MCF-7/HRG cells to anchorage-independently grow in soft-agar. In vivo treatment with a FASN inhibitor restored the anti-tumor activity of tamoxifen and fulvestrant against fast-growing, hormone-resistant MCF-7/HRG xenograft tumors in mice. Overall, these findings implicate FASN as a key enabler for endocrine resistance in HRG+/HER2- breast cancer and highlight the therapeutic potential of FASN inhibitors for the treatment of endocrine therapy-resistant luminal-B breast cancer.

## 1. Introduction

Patients with estrogen receptor (ER)-positive (ER+) breast cancer can benefit from long-term endocrine treatment. However, specific tumor phenotypic traits are important in determining the prognosis of women with ER+ breast cancer undergoing treatment with selective ER modulators (SERMs) and down-regulators (SERDs) such as tamoxifen and fulvestrant (formerly ICI 182,780), respectively [1,2,3,4,5]. In this context, the expression levels of proliferation-related genes can define two clinically distinct molecular subtypes of ER+ breast carcinoma—low-proliferative luminal-A and high-proliferative luminal-B—that differ in treatment response and clinical outcome [6,7,8,9,10]. When treated with tamoxifen, patients with luminal A-like tumors have a better prognosis than those with luminal B-like tumors. 

Investigations into the biological basis for these clinical observations by gene set enrichment analyses revealed that, independently of HER2 overexpression, growth factor signaling is significantly enriched in luminal B-tumors [11]. Specifically, ligand-induced transactivation of HER2 or other members of the epidermal growth factor receptor family in tumors with low levels of HER2 generates a gene signature strongly overlapping with that of the poor prognosis, luminal B-type of ER+ breast carcinomas, suggesting that HER-activating growth factors are significant contributors to the endocrine therapy-resistant phenotype. Accordingly, luminal A MCF-7 breast cancer cells, which are normally highly sensitive to tamoxifen in vitro, can overcome the anti-proliferative effects of tamoxifen when exogenously treated with the HER3 ligand heregulin (HRG) by enriching for the growth factor signaling gene set and activating a proliferative signature similar to that of tamoxifen-resistant ER+ luminal B-type tumors [11]. Further, previous studies from our laboratory revealed that transfection of tamoxifen-sensitive MCF-7 cells with an HRGβ-2 cDNA triggers the persistent activation of the HER2/3/4 receptors, and cells become estrogen-independent and resistant to anti-estrogens both in vitro and in vivo [12,13,14]. Beyond the expected hyperactivation of downstream signaling pathways such as ERK/MAPK and PI-3K/AKT, there remains a paucity of research into identifying prominent molecular targets that could be of therapeutic benefit in tamoxifen-refractory HRG+/HER2− luminal B-like breast carcinoma. 

Recent studies have revealed that lipid metabolism-related traits can be key drivers of breast cancer resistance to endocrine therapies including tamoxifen and aromatase inhibitors. For example, transcriptional profiling analysis followed by candidate gene expression and functional studies in long-term estrogen-deprived variant breast cancer cell lines has identified shared activation of sterol regulatory element-binding protein 1 (SREBP1) and of several SREBP1 downstream targets involved in fatty acid synthesis, including fatty acid synthase (FASN) [15]. Indeed, a significant association has been observed between the increase of SREBP1 expression in clinical specimens and the lack of clinical response to neo-adjuvant endocrine therapy, providing support for a role of SREBP1-related lipogenic programs in endocrine resistance in breast cancer [15]. The splicing factor epithelial splicing regulatory protein 1 has been shown to promote endocrine resistance and confer poor prognosis to patients with ER+ breast carcinoma by affecting lipid metabolism including the expression of FASN [16]. In this line, we recently reported the specific ability of FASN signaling to regulate the degree of sensitivity of breast cancer cells to estrogen-stimulated breast cancer cell growth and survival [17]. 

Here, using MCF-7 cells engineered to stably overexpress HRG as an experimental model of estrogen-independent cancer cell growth and endocrine resistance, we evaluated the autocrine capacity of HRG-driven HER2/HER3 signaling to stimulate FASN expression as part of the endocrine resistance program that is activated in certain subgroups of ER+ breast carcinomas. We present evidence that FASN is a key enabler for endocrine resistance in HRG+/HER2− breast cancer and highlight the therapeutic potential of FASN inhibitors for the treatment of endocrine therapy-resistant luminal B-like breast cancer.

## 2. Results

### 2.1. Heregulin Up-Regulates FASN Gene Expression by Autocrine Transactivation of HER2 Signaling

We first evaluated whether HRG overexpression and autocrine transactivation of HER2, independently of HER2 overexpression, might lead to the up-regulation of tumor-associated FASN in an in vitro model of endocrine resistant, luminal B-like breast cancer (Figure 1A, top panel). We evaluated FASN protein expression in an HRG-overexpressing model of biologically aggressive, endocrine-resistant ER+ breast cancer developed in our laboratory by transducing MCF-7 breast cancer cells with a retroviral vector containing the HRG cDNA (MCF-7/HRG cells [18]). To validate the autocrine requirement of HRG on HER2 signaling to activate FASN expression in the context of endocrine resistance, we re-evaluated FASN protein levels in MCF-7 cells engineered to overexpress an HRG structural deletion mutant incapable of promoting tumorigenicity and, more importantly for this study, incapable of establishing endocrine resistance [19]. The HRG-M4 deletion mutant lacks N-terminal sequences including a putative nuclear localization signal and the cytoplasmic-membrane domain (Figure 1A, bottom panel). Consequently, HRG-M4 protein is sequestered in the cellular compartment and cannot be secreted into the extracellular milieu, thereby preventing the autocrine activating action of HRG on HER receptors and the downstream activation of PI-3K/AKT and MAPK ERK1/2 signaling [18,19].

When the cellular pattern of FASN protein expression was assessed by indirect immunofluorescence microscopy, it was apparent and highly reproducible that cytoplasmic accumulation of FASN was notably higher in MCF-7/HRG cells than in MCF-7/pBABE control cells (Figure 1B, left microphotographs). In comparison with the pronounced cytoplasmic accumulation of FASN in MCF-7/HRG cells, immunofluorescence analyses suggested that MCF-7/HRG-M4 cells showed similar levels of FASN protein to those of HRG-negative MCF-7/pBABE control cells (Figure 1B, left microphotographs). When FASN fluorescence was semi-quantitatively analyzed by densitometry, the data indicated that FASN protein level was approximately 2.5-fold higher in MCF-7/HRG cells than in MCF-7/pBABE control cells. MCF-7/HRG-M4 cells, however, failed to up-regulate FASN protein expression (Figure 1B, middle panel).

To test whether HRG overexpression impacted *FASN* gene transcription downstream of HER2 transactivation, we transfected MCF-7/pBABE, MCF-7/HRG and MCF-7/HRG-M4 cells with a reporter construct containing a 178-bp *FASN* promoter fragment harboring all the elements necessary for high-level expression of FASN, including a complex SREBP-binding site [22,23,24]. *FASN* promoter-luciferase activity in MCF-7/HRG cells was significantly increased (~3-fold) relative to baseline levels in MCF-7/pBABE cells (Figure 1B, right panel). By contrast, *FASN* promoter-luciferase activity in MCF-7/HRG-M4 cells was equivalent to that observed in HRG-negative MCF-7/pBABE cells (Figure 1B, right panel).

### 2.2. PI-3K/AKT and MAPK ERK1/2 Signaling Pathways Drive Heregulin-Stimulated, SBREP-Dependent FASN Gene Expression in Breast Cancer Cells

To further corroborate that HRG-driven HER2 transactivation is involved in FASN accumulation in HRG-overexpressing MCF-7 cells, we employed a monoclonal antibody directed against the extracellular domain of HER2 (trastuzumab) that has been shown to inhibit HRG-induced HER2/3 phosphorylation and fully deactivate HER2-driven PI-3K/AKT and MAPK signaling in MCF-7/HRG cells [18]. Exposure to trastuzumab notably suppressed the hyperactive status of the *FASN* reporter construct in MCF-7/HRG cells (Figure 2A).

We next tested the contribution of the PI-3K/AKT signaling pathway to the FASN-overexpressing phenotype in MCF-7/HRG cells. Previous studies from our laboratory demonstrated that active AKT was significantly higher in MCF-7/HRG cells than in matched control MCF-7/pBABE cells, while the level of total AKT remained unchanged; a similar result was found for active MAPK [18]. Treatment with LY294002, a cell permeable inhibitor of PI-3K capable of suppressing hyperactive AKT signaling without affecting the active MAPK status in MCF-7/HRG cells [18], drastically decreased the activity of the *FASN* reporter in MCF-7/HRG cells to levels similar to those in MCF-7/pBABE cells (Figure 2B). We also investigated whether interfering with the MAPK signaling pathway prevented HRG-driven up-regulation of *FASN* by using U0126, a potent inhibitor of the MAPK-ERK1/2 pathway that was found to exclusively and completely suppress phospho-MAPK without affecting the status of phospho-AKT in MCF-7/HRG cells [18]. U0126 suppressed the stimulatory effects of HRG on *FASN* reporter activity, which returned to baseline control levels (Figure 2C). When MCF-7/HRG cells were transiently transfected with a *FASN* gene promoter construct with a deleted SREBP-binding region, HRG-driven stimulation of the *FASN* reporter was completely abolished (Figure 2D).

### 2.3. Pharmacological Blockade of FASN Activity Reverses HRG-Promoted Estrogen Independence and Endocrine Therapy-Resistance In Vitro

Because MCF-7 cells overexpressing HRG are estradiol-independent and acquire an anti-estrogen therapy-resistant phenotype [12,13,14], we explored whether exacerbated FASN activity might serve as part of the HRG-driven endocrine resistance program in breast cancer cells. To do this, we first measured anchorage-independent growth as an in vitro metric of tumorigenicity in response to estradiol and/or tamoxifen and fulvestrant. We found that MCF-7/pBABE control cells failed to form colonies in soft-agar in the absence of estradiol, whereas MCF-7/HRG cells showed a strong anchorage-independent capacity to form colonies (Figure 3A). The estrogen-independent tumorigenic phenotype of MCF-7/HRG cells was fully suppressed in a dose-dependent manner by treatment with the mycotoxin cerulenin or its semi-synthetic derivative C75, which are two widely employed small-molecule FASN inhibitors (Figure 3A).

As anticipated, exposure of MCF-7/pBABE cells to estradiol induced robust anchorage-independent growth, and this was prevented by tamoxifen (Figure 3B) and fulvestrant (Figure 3C). Estradiol treatment failed to further stimulate anchorage-independent colony formation in MCF-7/HRG cells and neither tamoxifen nor fulvestrant prevented the strong colony formation capacity of MCF-7/HRG cells. By contrast, when used in combination with tamoxifen, cerulenin and C75 dose-dependently suppressed the ability of MCF-7/HRG cells to form anchorage-independent colonies. Such sensitizing effects of the FASN blockers were more evident when cerulenin and C75 were combined with the pure anti-estrogen fulvestrant (Figure 3C), which antagonizes the hormone-dependent activation of ER but lacks the mixed antagonist/agonist effects of tamoxifen.

### 2.4. FASN Inhibition Reverses HRG-Mediated Resistance to Tamoxifen and Fulvestrant In Vivo

We finally determined whether pharmacological inhibition of FASN activity might overcome HRG-determined resistance to tamoxifen and fulvestrant in animal models. Ovariectomized nude mice were transplanted subcutaneously with endocrine-responsive MCF-7/pBABE cells and tamoxifen/fulvestrant-refractory MCF-7/HRG counterparts, and then were randomized into five groups (vehicle, tamoxifen, fulvestrant, C75, tamoxifen plus C75, and fulvestrant plus C75) following estrogen withdrawal (Figure 4A) or with continued estrogen supplementation (Figure 4B).

On day 50 after inoculation of cells, treatments with tamoxifen and fulvestrant were found to significantly prevent estrogen-stimulated tumor growth of MCF-7/pBABE control cells; treatment with C75 was not as efficient as tamoxifen and fulvestrant at preventing estrogen-driven MCF-7/pBABE tumor growth and failed to enhance the anti-estrogenic efficacy of tamoxifen and fulvestrant (Figure 4A). Estrogen supplementation was not required to establish MCF-7/HRG xenograft tumors in mice, thereby indicating a bona fide estrogen independency. The fast-growing pattern of MCF-7/HRG tumors remained unaltered upon treatment with tamoxifen and fulvestrant regardless estrogen withdrawal or estrogen supplementation. Treatment with the FASN inhibitor C75 drastically decreased the hormone-independent growth of MCF-7/HRG xenograft tumors, particularly when combined with tamoxifen and fulvestrant upon estrogen supplementation (Figure 4B). Taken together, these results provide evidence for FASN activity as a biological determinant that enables HRG-driven hormone-independence and refractoriness to SERMs (tamoxifen) and SERDs (fulvestrant) in ER+ breast cancer cells (Figure 4C).

## 3. Discussion

FASN is known to be differentially upregulated by the HER2 oncogene in breast epithelial cells [25,26,27,28]. By triggering PI-3K/AKT and MAPK signaling pathways, HER2 overexpression activates the FASN gene promoter and ultimately stimulates endogenous fatty acid synthesis. Indeed, the HER2-driven lipogenic phenotype might represent a biomarker for the sensitivity of pharmacological FASN blockade [20]. Here, we provide evidence that overexpression of the HER3 ligand HRG suffices to up-regulate the expression of cancer-associated FASN independently of HER2 overexpression. These findings confirm and extend earlier studies showing that the formation of a hyperactive heterodimer between HER2 and HER3 and the downstream activation of PI-3K/AKT and MAPK-ERK1/2 signaling cascades are essential traits for HRG-mediated elevation of FASN in breast cancer cells [29]. Perhaps more importantly, we now show that FASN is a key biological determinant of the molecular program through which HRG overexpression promotes the acquisition of an endocrine-resistant phenotype in ER+/HER2− breast cancer cells.

Activation of growth factor signaling pathways, independently of HER2 overexpression, has been suggested to contribute to the poor prognosis of the luminal B ER+ breast cancer subtype [11]. Our study provides evidence that activation of FASN signaling might operate as one of the dysfunctional biological pathways driving the highly proliferative [30], tamoxifen-resistant, phenotype of those tumors. This is not surprising, as many downstream growth factors activate PI-3K/AKT and MAPK-ERK1/2 signaling, and FASN activation therefore overlaps with phenotypic traits observed in breast cancer cell lines that overexpress HER2 [25,26,27,28,29]. In the present study, we utilized a structural mutant of HRG that, despite containing the EGF-like domain associated with most of the biological outcomes of ligand engagement with HER receptors, cannot be secreted to the extracellular milieu, thereby preventing the autocrine activating effects on HER2 [21]. Overexpression of the HRG structural mutant, which prevents the ability of full-length HRG to promote estradiol independence and antiestrogen resistance [18,19], failed to trigger PI-3K/AKT and MAPK-ERK1/2 signaling to transactivate the FASN gene. Crucially, our study might help to validate FASN as a predominant therapeutic target in certain subgroups of poor prognosis luminal-type breast carcinomas that would not require the inhibition of multiple pathways to produce clinical benefit. Accordingly, therapeutic blockade of FASN activity may be effective in the prevention of refractoriness to endocrine therapy in HER2-negative breast carcinomas subtypes where FASN overexpression results from HER2/HER3 signaling.

Endocrine-resistant MCF-7/HRG cells, which mimic a biological scenario of persistent HER2 pathway activation in luminal B-like tumors with low HER2 levels, lose their strong ability to form colonies in soft-agar under estrogen-depleted and/or anti-estrogen (tamoxifen and fulvestrant)-containing conditions in response to pharmacological blockade of FASN activity. The exact reason why FASN facilitates resistance to endocrine therapy in HRG-overexpressing, ER+/HER2− luminal B-like breast cancer cells needs to be explored in depth; however, it is clear that FASN pro-survival signaling [27,28,31] is co-opted to support the cross-talk between HRG-driven HER2/HER3 pathway activation and ER signaling, thereby enabling estrogen-independence and resistance to SERMs/SERDs (Figure 5). Importantly, in vivo treatment with the FASN inhibitor C75 notably restored the anti-tumor activity of tamoxifen and fulvestrant against fast-growing, hormone-resistant MCF-7/HRG xenograft tumors in mice. The capacity of anti-FASN therapy to restore the sensitivity to tamoxifen and fulvestrant in MCF-7/HRG xenograft tumors certainly suggests that certain subgroups of HRG-overexpressing, ER-positive/HER2-negative patients who phenotypically behave as triple-negative breast carcinomas and are resistant to endocrine therapy could greatly benefit from adding clinical-grade FASN inhibitors [31,32,33] to combined treatments with SERMs/SERDs.

## 4. Materials and Methods

### 4.1. Materials

LY294002 and U0126 (Calbiochem, San Diego, CA, USA) were dissolved in DMSO and stored in the dark as stock solutions (10 mmol/L) at −20 °C until use. Control cells were cultured in medium containing the same concentration of DMSO (*v*/*v*) as was used in treated samples. Trastuzumab (Herceptin^®^) was solubilized in bacteriostatic water for injection containing 1.1% benzyl alcohol (stock solution 21 mg/mL), stored at 4 °C and used within one month. The primary antibody for FASN immunostaining was a mouse IgG_1_ FASN monoclonal antibody (clone 23) from BD Biosciences Pharmingen (San Jose, CA, USA).

### 4.2. Heregulin Constructs and Retroviral Infection of Cell Lines

The HRG-M4 deletion mutant was generated by PCR using the HRGβ2 cDNA (accession number 183996) as a template, as described [19]. Full-length HRGβ2 cDNA and HRG-M4 cDNA were cloned into the retroviral vector pBABE (kindly provided by J. Campisi, Lawrence Berkeley National Laboratory, University of California, Berkeley, CA, USA), and each was transfected into a high efficiency transient packaging system using FuGENE (Roche Biochemicals, Indianapolis, IN, USA). Medium from transfected cells containing infectious retrovirus was collected after 48 h, filtered, and used to infect MCF-7 cells (originally obtained from the American Type Culture Collection, Manassas, WV, USA) for 24 h in the presence of 4 μg/mL polybrene (Sigma, St. Luis, MO, USA). Infected MCF-7 cells were grown for an additional 24 h in standard medium and stable cell lines (MCF-7/pBABE, MCF-7/HRG and MCF-7/HRG-M4) were selected and expanded in the presence of 2.5 μg/mL puromycin (Sigma). MCF-7/pBABE, MCF-7/HRG and MCF-7/HRG-M4 cells were routinely maintained in phenol red-containing improved MEM (Biosource International, Camarillo, CA, USA) containing 5% (*v*/*v*)-heat-inactivated fetal bovine serum and 2 mmol/L l-glutamine. Cells were maintained at 37 °C in a humidified atmosphere of 95% air and 5% CO_2_.

### 4.3. Immunofluorescence Staining

Anti-FASN immunofluorescence was performed as described elsewhere [20]. Fluorescence intensities from regions of interest (ROIs) representing each cell were semi-quantitatively analyzed by densitometry (Image J software, which can be readily downloaded from the NIH website https://imagej.nih.gov/ij/download.html), and the individual intensity values were employed to derive an average intensity for all the cells in the image.

### 4.4. FASN Promoter Activity

Cells were trypsinized and re-plated in 24-well plates at a density of 50,000 cells/well. Cells were incubated for 18 h to allow for attachment and were then transfected with 300 ng/well of the pGL3-Luciferase construct (Promega, Madison, WI, USA) containing a Luciferase reporter gene driven by a 178-bp *FASN* promoter fragment using FuGENE 6 transfection reagent (Roche Biochemicals). An internal control plasmid pRL-CMV (30 ng/well) was used to correct for transfection efficiency. After 18 h, transfected cells were washed and then incubated with trastuzumab, LY294002, or U0126 as specified, or vehicles as controls. Approximately 24 h after treatments, Luciferase assays were performed as previously described [19,20].

### 4.5. Xenograft Studies

Animal care was in accordance with institutional guidelines. Xenografts were established by injecting 2 × 10^6^ (MCF-7/pBABE and MCF-7/HRG) cells subcutaneously into ovariectomized 3- to- 4-week old athymic female nude-Doxn1^nu^ mice (Harlan Sprague Dawley, Madison, WI, USA) that had been implanted with slow-release estrogen pellets (Innovative Research) implanted subcutaneously around left forearm using a trocar. When tumors reached a size of approximately 100 mm^3^, mice bearing MCF-7/pBABE and MCF-7/HRG tumors were randomly allocated to continued estrogen treatment or to estrogen withdrawal (by removal of the estrogen pellets) with vehicle alone (untreated group), single-agent tamoxifen, single-agent fulvestrant, single-agent C75 (30 mg/Kg b.w.; i.p.), or their combinations (i.e., tamoxifen plus C75 or fulvestrant plus C75) for seven weeks (*n* = 10 animals/experimental group). Tumor volume was calculated by 3D measurements using the formula: tumor volume (mm^3^) = (length × width × height)/2. Tumor volume values (mean ± S.D.) were calculated using a Vernier caliper in a blinded manner to minimize experimental bias. Mice were euthanized at completion of the experiment (50 days post-inoculation) or when tumors reached a volume of 1000 mm^3^, and tumor tissues were removed and maintained at −190 °C for later analyses. Animal studies were conducted in accordance with the Guide for the Care and Use of Laboratory Animals and the Mayo Clinic Institutional Animal Care and Use Committee (IACUC) approved this study (protocol numbers A40611-12, 15 February 2012 -initial approval- and A00004567-19, 16 December 2019 -last approval).

### 4.6. Statistical Analysis

For all experiments, at least three independent experiments were performed with *n* ≥ 3 replicate samples per experiment. Investigators were blinded to animal data allocation. Experiments were not randomized. Data are presented as mean ± S.D. Comparisons of means of ≥3 groups were performed by one-way ANOVA and Dunnett’s t-test for multiple comparisons using GraphPad Prism (GraphPad Software, San Diego, CA, USA). In all studies, *p*-values < 0.05 and <0.005 were considered to be statistically significant (denoted as * and **, respectively). All statistical tests were two-sided.

## 5. Conclusions

With the exception of the HER2 subtype, there has been little research into the pathogenic mechanisms responsible for the intrinsic molecular subtypes of breast cancer. Elucidation of the underlying biological mechanisms contributing to the luminal-B/poor prognostic ER+ breast cancer phenotype is critical for developing novel and effective therapeutic strategies aimed to circumvent endocrine resistance. We propose that lipid-metabolic traits such as FASN, which is becoming increasingly important as a potential biomarker of poor prognosis and therapeutic target in several human malignancies [19,34,35,36,37], contribute to the highly proliferative, hormone therapy-insensitive phenotype of luminal breast cancer carcinomas in a HER2 overexpression-independent manner. As a new generation of FASN inhibitors has recently entered the clinic, our study suggests that targeting FASN could be therapeutically exploited for the clinico-molecular management of the poor prognosis luminal-B subtype of ER+ breast cancer patients.

## Figures and Tables

**Figure 1 ijms-21-07661-f001:**
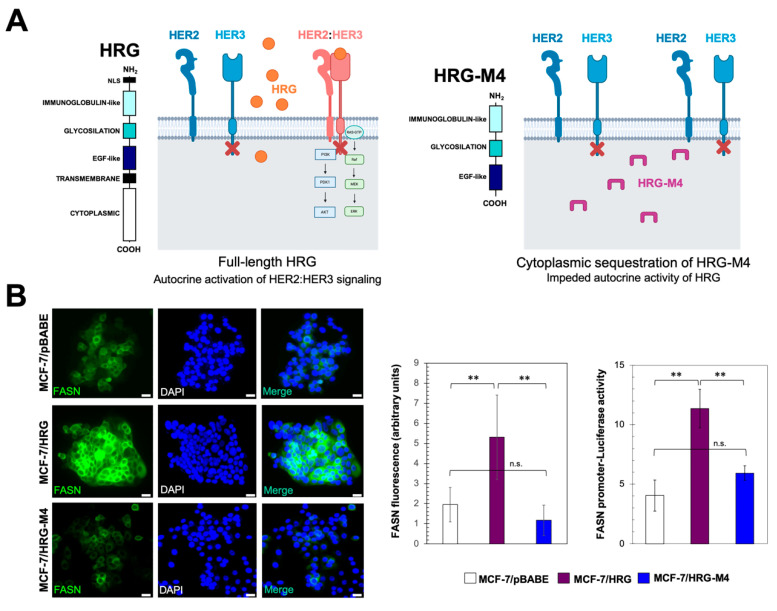
Luminal breast cancer cells engineered to overexpress the HER3 ligand heregulin up-regulate FASN expression. (**A**) The structural deletion mutant HRG-M4 suffers a cytoplasmic sequestration that impedes the natural autocrine capacity of full-length HRG to promote the trans-phosphorylation of the HER3 receptor within HER2:HER3 heterodimers. (**B**) Left microphotographs. MCF-7/pBABE, MCF-7/HRG, and MCF-7/HRG-M4 cells were subjected to immunofluorescence staining with an anti-FASN specific antibody as described in “Materials and methods”. *Middle panel*. FASN fluorescence intensities in MCF-7/pBABE, MCF-7/HRG, and MCF-7/HRG-M4 cells were semi-quantitively analyzed by densitometry. Each experimental value represents the mean FASN fluorescence (arbitrary units, columns) ± S.D. (bars) for all the cells in the images on the left. Right panel. MCF-7/pBABE, MCF-7/HRG, and MCF-7/HRG-M4 cells were transiently transfected with a plasmid containing a Luciferase reporter gene driven by a 178-bp *FASN* gene promoter fragment harboring a SREBP-binding site, flanked by auxiliary NF-Y and Sp-1 sites (Figure 2). After 48 h, luciferase assays were performed as previously described [20,21]. Each experimental value represents the mean fold-increase (columns) ± S.D. (bars) from at least three separate experiments in which triplicate wells were measured. ** *p* < 0.005; n.s. not significant. (Scale bar is 10 µm).

**Figure 2 ijms-21-07661-f002:**
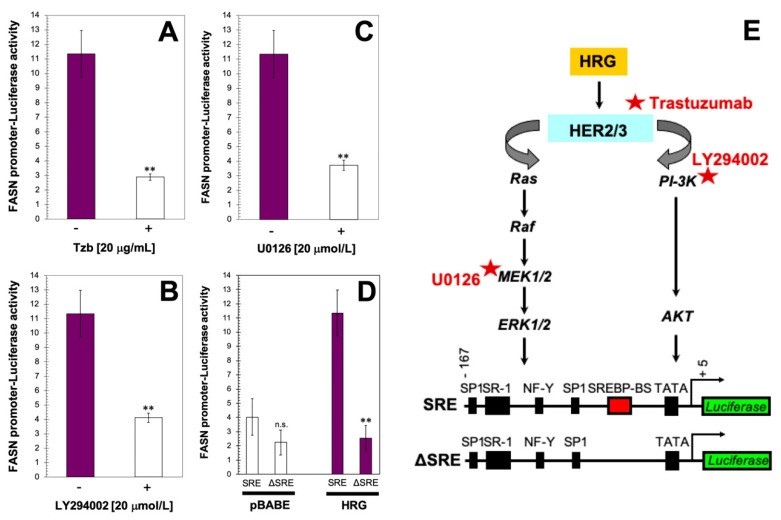
PI-3K/AKT and MAPK ERK1/2 signaling cascades mediate heregulin-driven activation of *FASN* gene expression in a SBREP-dependent manner. MCF-7/HRG cells were transiently transfected with a plasmid containing a Luciferase reporter gene driven by a 178-bp *FASN* gene promoter fragment harboring a SREBP-binding site, flanked by auxiliary NF-Y and Sp-1 sites. The next day, cells were treated with 20 μg/mL of trastuzumab (Tzb) (**A**), LY294002 (**B**), or U0126 (**C**). Alternatively, MCF-7/pBABE and MCF-7/HRG cells were transiently transfected with a plasmid containing a Luciferase reporter gene driven by a 178-bp *FASN* promoter fragment harboring a SREBP-binding site or with a similar construct with the SREBP-binding site deleted (**D**). After 24 h, cells were lysed and Luciferase activity was measured. Luciferase assays were performed as previously described [19,20]. Each experimental value represents the mean fold increase (columns) ± S.D. (bars) from at least three separate experiments in which triplicate wells were measured. ** *p* < 0.005; n.s. not statistically significant. (**E**) HRG-triggered regulatory cascade actively links upstream HER2 transactivation and increased activation of PI-3K/AKT-MAPK/ERK1/2 signaling transduction with downstream (SREBP-dependent) transcriptional activation of the *FASN* gene promoter.

**Figure 3 ijms-21-07661-f003:**
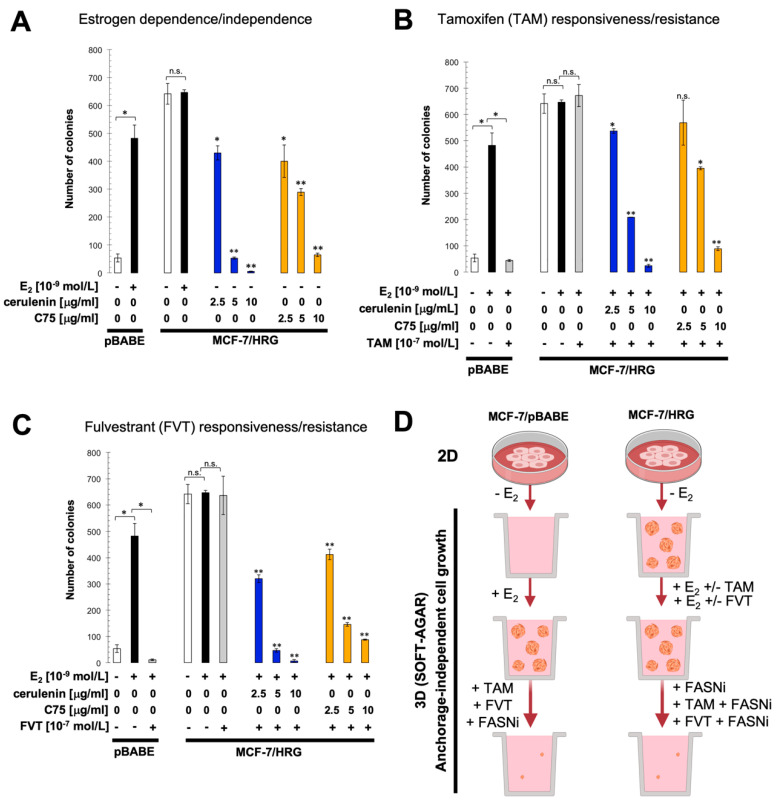
Pharmacological inhibition of FASN activity impedes estrogen-independent and antiestrogen-refractory cell growth of MCF-7/HRG cells. Estradiol (E_2_)-depleted cells were plated in soft agarose containing E_2_ (10^−9^ mol/L) (**A**), cerulenin (2.5, 5, and 10 μg/mL), C75 (2.5, 5, and 10 μg/mL), tamoxifen (10^−7^ mol/L) (**B**), fulvestrant (10^−7^ mol/L) (**C**), their combinations, or ethanol (*v*/*v*) or DMSO (*v*/*v*) vehicle only for 7–10 days. Colony formation (≥50 μm) was assessed using a colony counter. Each experimental value represents the mean colony number (columns) ± S.D. (bars) from at least three separate experiments in which triplicate dishes were counted. (* *p* < 0.05; ** *p* < 0.005; n.s. not statistically significant). (**D**) Pharmacological inhibition of FASN activity blocked the estradiol-independent and tamoxifen/fulvestrant-refractory ability of MCF-7/HRG cells to anchorage-independently grow in soft-agar. FASNi: FASN inhibitor; TAM: Tamoxifen; FVT: Fulvestrant;º 2D: Two-dimensional; 3D: Three-dimensional.

**Figure 4 ijms-21-07661-f004:**
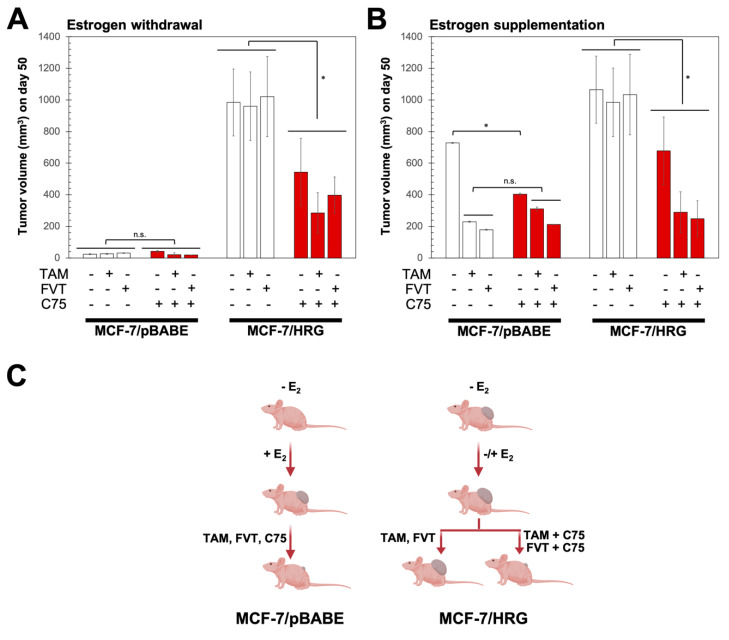
FASN inhibition restores anti-tumor activity of tamoxifen and fulvestrant against MCF-7/HRG tumors. Mean tumor volumes (mm^3^) (columns) ± S.D. (bars) on day 50 of MCF-7/pBABE and MCF-7/HRG xenograft tumors in athymic female mice treated with the FASN inhibitor C75 in the absence (**A**) or presence (**B**) of estrogen, tamoxifen (TAM), and/or fulvestrant (FVT) (*n* = 10 mice/experimental group). (**C**) In vivo treatment with the FASN inhibitor C75 restores the anti-tumor activity of TAM and FVT against hormone-resistant MCF-7/HRG xenograft tumors in mice.

**Figure 5 ijms-21-07661-f005:**
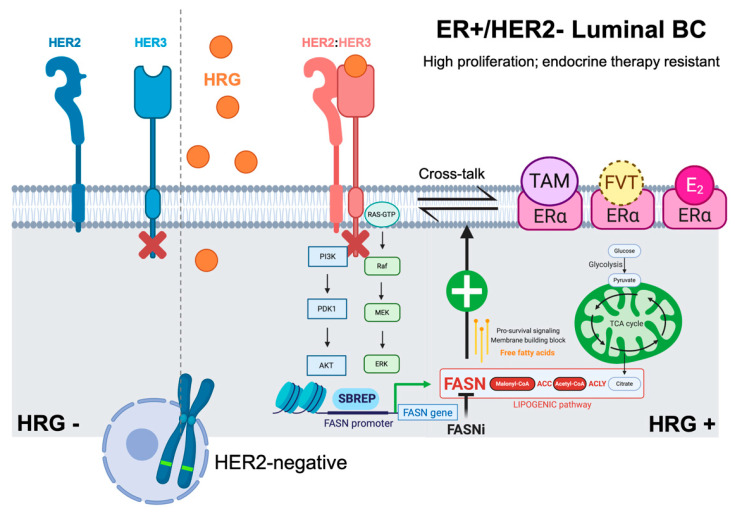
FASN is part of the endocrine resistance program in HRG-positive/HER2-negative breast cancer cells. The most predominant mechanisms of endocrine therapy resistance in estrogen receptor-positive (ER+) breast cancer include interaction between ER signaling and various growth factor pathways such as HRG-driven transactivation of HER signaling in the absence of HER2 overexpression. These alterations facilitate adaptation from estradiol-dependent to estradiol-independent ER activation, which is further triggered by cross-talk with growth factor receptors (e.g., HRG-driven HER2:HER3 heterodimerization). We now provide evidence that FASN is a key mediator in promoting HRG-driven endocrine-resistant breast cancer.

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
