# Peer review of "Fatty Acid Synthase Is a Key Enabler for Endocrine Resistance in Heregulin-Overexpressing Luminal B-Like Breast Cancer"

_ijms, 2020, doi:10.3390/ijms21207661_

Round 1

Reviewer 1 Report

Javier A. Menendez and collaborators wrote a Brief Report regarding lipogenic enzyme fatty acid synthase (FASN) as a key enabler for endocrine resistance in HRG+/HER2- breast cancer and highlight the therapeutic potential of FASN 51 inhibitors for the treatment of endocrine therapy-resistant luminal-B breast cancer. The report is well-written, but I have a few suggestions:

  • Figure 1B shows immunofluorescence for FASN. It would be interesting to add a merge image (DAPI+ FASN) and a quantitative analysis.
  • The Authors should check the manuscript carefully for acronyms (they should appear at the first citation in the text and in the Abbreviations section)

Author Response

REVIEWER #1

Javier A. Menendez and collaborators wrote a Brief Report regarding lipogenic enzyme fatty acid synthase (FASN) as a key enabler for endocrine resistance in HRG+/HER2- breast cancer and highlight the therapeutic potential of FASN 51 inhibitors for the treatment of endocrine therapy-resistant luminal-B breast cancer. The report is well-written, but I have a few suggestions:

  • Figure 1B shows immunofluorescence for FASN. It would be interesting to add a merge image (DAPI+ FASN) and a quantitative analysis.

Reply: Figure 1 has been redesigned and updated to merge FASN and DAPI as well as to provide a semi-quantitative analysis of FASN immunofluorescence, as per the reviewer’s suggestion.

Page 4, lines 122-154

Figure 1. Luminal breast cancer cells engineered to overexpress the HER3 ligand heregulin up-regulate FASN expression. A. The structural deletion mutant HRG-M4 suffers a cytoplasmic sequestration that impedes the natural autocrine capacity of full-length HRG to promote the trans-phosphorylation of the HER3 receptor within HER2:HER3 heterodimers. B. Left microphotographs. MCF-7/pBABE, MCF-7/HRG, and MCF-7/HRG-M4 cells were subjected to immunofluorescence staining with an anti-FASN specific antibody as described in “Materials and methods”. Middle panel. FASN fluorescence intensities in MCF-7/pBABE, MCF-7/HRG, and MCF-7/HRG-M4 cells were semi-quantitively analyzed by densitometry. Each experimental value represents the mean FASN fluorescence (arbitrary units, columns) ± S.D. (bars) for all the cells in the images on the left. Right panel. MCF-7/pBABE, MCF-7/HRG, and MCF-7/HRG-M4 cells were transiently transfected with a plasmid containing a Luciferase reporter gene driven by a 178-bp FASN gene promoter fragment harboring a SREBP-binding site, flanked by auxiliary NF-Y and Sp-1 sites (see Figure 2). After 48 h, luciferase assays were performed as previously described [19,20]. Each experimental value represents the mean fold-increase (columns) ± S.D. (bars) from at least three separate experiments in which triplicate wells were measured. **P < 0.005; n.s. not significant.

When the cellular pattern of FASN protein expression was assessed by indirect immunofluorescence microscopy, it was apparent and highly reproducible that cytoplasmic accumulation of FASN was notably higher in MCF-7/HRG cells than in MCF-7/pBABE control cells (Figure 1B, left microphotographs). In comparison with the pronounced cytoplasmic accumulation of FASN in MCF-7/HRG cells, immunofluorescence analyses suggested that MCF-7/HRG-M4 cells showed similar levels of FASN protein to those of HRG-negative MCF-7/pBABE control cells (Figure 1B, left microphotographs). When FASN fluorescence was semi-quantitatively analyzed by densitometry, the data indicated that FASN protein level was approximately 2.5-fold higher in MCF-7/HRG cells than in MCF-7/pBABE control cells. MCF-7/HRG-M4 cells, however, failed to up-regulate FASN protein expression (Figure 1B, middle panel).

To test whether HRG overexpression impacted FASN gene transcription downstream of HER2 transactivation, we transfected MCF-7/pBABE, MCF-7/HRG and MCF-7/HRG-M4 cells with a reporter construct containing a 178-bp FASN promoter fragment harboring all the elements necessary for high-level expression of FASN, including a complex SREBP-binding site [22-24]. FASN promoter-luciferase activity in MCF-7/HRG cells was significantly increased (~3-fold) relative to baseline levels in MCF-7/pBABE cells (Figure 1B, right panel). By contrast, FASN promoter-luciferase activity in MCF-7/HRG-M4 cells was equivalent to that observed in HRG-negative MCF-7/pBABE cells (Figure 1B, right panel).

Page 10, lines 327-332

4.3. Immunofluorescence staining

Anti-FASN immunofluorescence was performed as described elsewhere [20]. Fluorescence intensities from regions of interest (ROIs) representing each cell were semi-quantitatively analyzed by densitometry (Image J software, which can be readily downloaded from the NIH website https://imagej.nih.gov/ij/download.html), and the individual intensity values were employed to derive an average intensity for all the cells in the image.

  • The Authors should check the manuscript carefully for acronyms (they should appear at the first citation in the text and in the Abbreviations section)

Done!

Reviewer 2 Report

The manuscript by Menendez et al. ascertained the involvement of the lipogenic enzyme fatty acid synthase (FASN) in the heregulin (HRG) dependent endocrine resistance in ER+/HER2-negative breast cancer cells.  In this regard, the Authors determined that FASN is up-regulated in MCF-7 cells stably overexpressing HRG (MCF-7/HRG), which were used as model system of tamoxifen/fulvestrant-resistant luminal B-like breast cancer. FASN inhibition prevented the estradiol-independent ability of MCF-7/HRG cells to grow in soft-agar and the treatment with a FASN inhibitor re-established the activity of tamoxifen and fulvestrant in hormone-resistant MCF-7/HRG xenograft tumors in mice. On the basis of these results the authors suggested that FASN may be a useful targets in endocrine therapy-resistant luminal-B breast cancer.

The issue is interesting as it deals with novel findings on the potential of FASN to contribute to breast cancer progression. However, the discussion should include some other observations (i.e. https://doi.org/10.3390/ijms150711539 ; https://doi.org/10.3390/ijms21061931) in order to corroborate the role FSN in cancer development. In addition, it should be detected by an adequate software the IF of figure 1B.

Author Response

REVIEWER #2

The manuscript by Menendez et al. ascertained the involvement of the lipogenic enzyme fatty acid synthase (FASN) in the heregulin (HRG) dependent endocrine resistance in ER+/HER2-negative breast cancer cells.  In this regard, the Authors determined that FASN is up-regulated in MCF-7 cells stably overexpressing HRG (MCF-7/HRG), which were used as model system of tamoxifen/fulvestrant-resistant luminal B-like breast cancer. FASN inhibition prevented the estradiol-independent ability of MCF-7/HRG cells to grow in soft-agar and the treatment with a FASN inhibitor re-established the activity of tamoxifen and fulvestrant in hormone-resistant MCF-7/HRG xenograft tumors in mice. On the basis of these results the authors suggested that FASN may be a useful targets in endocrine therapy-resistant luminal-B breast cancer.

The issue is interesting as it deals with novel findings on the potential of FASN to contribute to breast cancer progression. However, the discussion should include some other observations (i.e. https://doi.org/10.3390/ijms150711539 https://doi.org/10.3390/ijms21061931) in order to corroborate the role FSN in cancer development.

Reply: We have now added few references corroborating the role of FASN in cancer development.

  1. Conclusions

With the exception of the HER2 subtype, there has been little research into the pathogenic mechanisms responsible for the intrinsic molecular subtypes of breast cancer. Elucidation of the underlying biological mechanisms contributing to the luminal-B/poor prognostic ER+ breast cancer phenotype is critical for developing novel and effective therapeutic strategies aimed to circumvent endocrine resistance. We propose that lipid-metabolic traits such as FASN, which is becoming increasingly important as a potential biomarker of poor prognosis and therapeutic target in several human malignancies [19, 34-37], contribute to the highly proliferative, hormone therapy-insensitive phenotype of luminal breast cancer carcinomas in a HER2 overexpression-independent manner. As a new generation of FASN inhibitors has recently entered the clinic, our study suggests that targeting FASN could be therapeutically exploited for the clinico-molecular management of the poor prognosis luminal-B subtype of ER+ breast cancer patients.

  1. Jiang L, Wang H, Li J, Fang X, Pan H, Yuan X, Zhang P. Up-regulated FASN expression promotes transcoelomic metastasis of ovarian cancer cell through epithelial-mesenchymal transition. J. Mol. Sci. 2014, 15, 11539-11554.
  2. Ricklefs, F.L.; Maire, C.L.; Matschke, J.; Dührsen, L.; Sauvigny, T.; Holz, M.; Kolbe, K.; Peine, S.; Herold-Mende, C.; Carter, B.; Chiocca, E.A.; Lawler, S.E.; Westphal, M.; Lamszus, K. FASN Is a Biomarker Enriched in Malignant Glioma-Derived Extracellular Vesicles. J. Mol. Sci. 2020, 21, 1931. 
  3. Gonzalez-Guerrico, A.M.; Espinoza, I.; Schroeder, B.; Park, C.H.; Kvp, C.M.; Khurana, A.; Corominas-Faja, B.; Cuyàs, E.; Alarcón, T.; Kleer, C.; Menendez, J.A.; Lupu, R. Suppression of endogenous lipogenesis induces reversion of the malignant phenotype and normalized differentiation in breast cancer. Oncotarget 2016, 7, 71151-71168. 
  4. Su, Y.W.; Wu, P.S.; Lin, S.H.; Huang, W.Y.; Kuo, Y.S.; Lin, H,P.Prognostic Value of the Overexpression of Fatty Acid Metabolism-Related Enzymes in Squamous Cell Carcinoma of the Head and Neck. J. Mol. Sci. 2020, 21, E6851.

In addition, it (FASN) should be detected by an adequate software the IF of figure 1B.

Figure 1 has been redesigned and updated to merge FASN and DAPI as well as to provide a semi-quantitative analysis of FASN immunofluorescence, as per the reviewer’s suggestion.

Page 4, lines 122-154

Figure 1. Luminal breast cancer cells engineered to overexpress the HER3 ligand heregulin up-regulate FASN expression. A. The structural deletion mutant HRG-M4 suffers a cytoplasmic sequestration that impedes the natural autocrine capacity of full-length HRG to promote the trans-phosphorylation of the HER3 receptor within HER2:HER3 heterodimers. B. Left microphotographs. MCF-7/pBABE, MCF-7/HRG, and MCF-7/HRG-M4 cells were subjected to immunofluorescence staining with an anti-FASN specific antibody as described in “Materials and methods”. Middle panel. FASN fluorescence intensities in MCF-7/pBABE, MCF-7/HRG, and MCF-7/HRG-M4 cells were semi-quantitively analyzed by densitometry. Each experimental value represents the mean FASN fluorescence (arbitrary units, columns) ± S.D. (bars) for all the cells in the images on the left. Right panel. MCF-7/pBABE, MCF-7/HRG, and MCF-7/HRG-M4 cells were transiently transfected with a plasmid containing a Luciferase reporter gene driven by a 178-bp FASN gene promoter fragment harboring a SREBP-binding site, flanked by auxiliary NF-Y and Sp-1 sites (see Figure 2). After 48 h, luciferase assays were performed as previously described [19,20]. Each experimental value represents the mean fold-increase (columns) ± S.D. (bars) from at least three separate experiments in which triplicate wells were measured. **P < 0.005; n.s. not significant.

When the cellular pattern of FASN protein expression was assessed by indirect immunofluorescence microscopy, it was apparent and highly reproducible that cytoplasmic accumulation of FASN was notably higher in MCF-7/HRG cells than in MCF-7/pBABE control cells (Figure 1B, left microphotographs). In comparison with the pronounced cytoplasmic accumulation of FASN in MCF-7/HRG cells, immunofluorescence analyses suggested that MCF-7/HRG-M4 cells showed similar levels of FASN protein to those of HRG-negative MCF-7/pBABE control cells (Figure 1B, left microphotographs). When FASN fluorescence was semi-quantitatively analyzed by densitometry, the data indicated that FASN protein level was approximately 2.5-fold higher in MCF-7/HRG cells than in MCF-7/pBABE control cells. MCF-7/HRG-M4 cells, however, failed to up-regulate FASN protein expression (Figure 1B, middle panel).

To test whether HRG overexpression impacted FASN gene transcription downstream of HER2 transactivation, we transfected MCF-7/pBABE, MCF-7/HRG and MCF-7/HRG-M4 cells with a reporter construct containing a 178-bp FASN promoter fragment harboring all the elements necessary for high-level expression of FASN, including a complex SREBP-binding site [22-24]. FASN promoter-luciferase activity in MCF-7/HRG cells was significantly increased (~3-fold) relative to baseline levels in MCF-7/pBABE cells (Figure 1B, right panel). By contrast, FASN promoter-luciferase activity in MCF-7/HRG-M4 cells was equivalent to that observed in HRG-negative MCF-7/pBABE cells (Figure 1B, right panel).

Page 10, lines 327-332

4.3. Immunofluorescence staining

Anti-FASN immunofluorescence was performed as described elsewhere [20]. Fluorescence intensities from regions of interest (ROIs) representing each cell were semi-quantitatively analyzed by densitometry (Image J software, which can be readily downloaded from the NIH website https://imagej.nih.gov/ij/download.html), and the individual intensity values were employed to derive an average intensity for all the cells in the image.